# Complete Left-Sided Pericardial Congenital Absence

**DOI:** 10.3390/reports7020048

**Published:** 2024-06-20

**Authors:** Petar Kalaydzhiev, Anelia Partenova, Radostina Ilieva, Kamelia Genova, Elena Kinova

**Affiliations:** 1Cardiology Department, University Hospital “Tsaritsa Yoanna—ISUL”, 1000 Sofia, Bulgaria; radostilieva@yahoo.com (R.I.);; 2Emergency Department, Medical University, 1000 Sofia, Bulgaria; 3MRI Centre, UMBALSM “N. I. Pirogov”, 1000 Sofia, Bulgaria; apartenova@gmail.com (A.P.);; 4St. Ekaterina University Hospital, 1000 Sofia, Bulgaria

**Keywords:** pericardium, congenital absence of pericardium, cardiac magnetic resonance, echocardiography

## Abstract

Background: Congenital absence of pericardium is a rare cardiac disorder with a reported incidence of less than 1 in 10,000. Although most of the cases are of little clinical significance, some of them are associated with serious complications, including risk of herniation and strangulation or coronary artery compression. Detailed Case Description: We report a case of a 36-year-old male referred for routine cardiovascular examination. He had a medical history of a heart murmur since childhood. Electrocardiogram (ECG) revealed sinus rhythm, normal axis, poor R-wave progression in the precordial leads and repolarization abnormalities with negative T waves in leads V1–V4. On 2D transthoracic echocardiography (TTE), an unusual heart position was noted with poor image quality from the standard acoustic windows. The parasternal long axis view gave the impression of right ventricular dilatation. The findings raised the suspicion of left to right shunt and possible atrial septal defect. For further evaluation, the patient was referred for cardiac magnetic resonance which demonstrated complete left-sided absence of the pericardium. Discussion: Due to indistinct and atypical symptoms and lack of clinical awareness, pericardial congenital absence is frequently misdiagnosed. Patients may complain of atypical chest pain. Patient’s history and physical examination are often nonspecific. In cases with complete pericardial absence, ECG findings may include right axis deviation, right bundle block and sinus bradycardia. Echocardiography findings are also not characteristic, but some may raise the clinical suspicion of this diagnosis. The imaging modalities of choice are computed tomography and cardiac magnetic resonance. Treatment depends on the type of defect and clinical symptoms.

## 1. Introduction

Congenital absence of pericardium (CAP) is a rare cardiac disorder with a reported incidence of less than 1 in 10,000 [1]. Depending on the defect’s size and its location, CPA can be divided into complete and partial. The most common type is a complete left-sided defect with a prevalence of 70% of all pericardial defects. It is more common in males. Right-sided defects are reported with an incidence of 17%, and the rarest form is the complete bilateral pericardial absence—9% of all defects [2,3]. The complete pericardial absence is usually of little clinical significance, and this condition is most commonly an incidental finding on imaging or at surgery for other reasons. In contrast, the partial pericardial defects are associated with more severe symptoms and sometimes life-threatening complications. Up to fifty percent of the cases with CAP are also associated with other congenital cardiac anomalies like an atrial septal defect, patent ductus arteriosus, tetralogy of Fallot, sinus venosus defect, and mitral valve disorders [4]. We report a case of complete left-sided pericardial agenesis, incidentally found in a 36-year-old man who was referred for cardiac magnetic resonance (CMR) for evaluation on an atrial septal defect.

## 2. Detailed Case Description

A 36-year-old male with first grade essential arterial hypertension with good medical control was referred for routine cardiovascular exam. He had a medical history of heart murmur since childhood and has been told to have interventricular communication. No prior medical records were submitted. Physical examination did not demonstrate any cardio-pulmonary abnormalities. An electrocardiogram (ECG) revealed sinus rhythm at 62 bpm, normal axis, poor R-wave progression in the precordial leads and repolarization abnormalities and negative T waves in V1–V4 (Figure 1).

On 2D transthoracic echocardiography (TTE), an unusual heart position was noted with poor image quality from the standard acoustic windows. The parasternal long axis view gave the impression of right ventricular dilatation. An exaggerated motion of the left ventricular posterior wall was noted with hyperechogenic space behind it, which initially has been thought to be pericardium. The left ventricular apex was displaced laterally and posteriorly and was impossible to be visualized from the standard apical views (Figure 2). The apex was better demonstrated when the patient was not in left lateral but in supine position. The findings raised the suspicion of the atrial septal defect, and the patient was referred for CMR for further evaluation.

The CMR confirmed the abnormal position of the heart in the left hemithorax with laterally and posteriorly displaced apex and interposition of the lung tissue between the aorta and pulmonary trunk. There was no detectable pericardium overlying the left ventricular wall, and only a small discontinuous segments of pericardium were visualized surrounding the right atrium and right ventricular lateral wall (Figure 3 and Figure 4). Biventricular volumes and function were in the reference range. Additionally, a patent foramen ovale was visualized with no significant left-to-right shunt (Qp/Qs = 1.1).

No other cardiac malformations were found. Taking into consideration all of the information, a diagnosis of complete left-sided pericardial agenesis with persistent foramen ovale was made. The patient was followed up after six months, with no complaints and no therapy.

## 3. Discussion

The etiology of pericardial agenesis is not fully understood but is thought to be due to the early atrophy of Cuvier’s ducts. They are involved in the development of the pleuro-pericardial membranes. Pericardial defects are more common on the left side, and this is probably due to the fact that the left Cuvier’s duct normally atrophies, and its premature obliteration causes left-sided defects. The right duct of Cuvier persists as the superior vena cava which is thought to assure the normal formation of the right pleuro-pericardial membrane, and thus, right-sided defects are much more uncommon [5].

Due to indistinct and atypical symptoms and lack of clinical awareness, pericardial agenesis is frequently misdiagnosed. Patients with bilateral or complete left-sided pericardial agenesis are most commonly asymptomatic but they may complain of atypical chest pain that is thought to be due to the kinking and strain of the great vessels [2]. It may be exaggerated when lying on the left side due to volume loading of the ventricles [6]. Patients with partial defects are more likely to have symptoms, again the most common among which is the atypical chest pain. The pain in these patients may be due to compression of the coronary arteries by the pericardial rim at the side of the defect [2,3]. This may mimic angina pectoris and should be considered in the differential diagnosis of these patients. Small foramen-type defects are of bigger risk for heart herniation and coronary artery compression. They may cause strangulation and necrosis of the heart, which requires surgical intervention [2].

In patients with CAP, the history and physical examination are often nonspecific and with no role in making the diagnosis. The ECG is usually normal in small or partial defects. In cases with complete pericardial absence, some typical findings include right axis deviation, complete or incomplete right bundle block and sinus bradycardia induced by vagal stimulation [7]. In these cases, chronic lung disease, congenital interventricular and interatrial septal defects with hemodynamically significant shunts should be considered. Echocardiography is also not characteristic, but some findings may raise the clinical suspicion of this diagnosis. They include unusual acoustic windows due to the abnormal heart position, right ventricle dilatation, cardiac hypermobility, “teardrop” appearance, and paradoxical motion of the interventricular septum [8]. In our case, a dilated right ventricle was registered at TTE, and in the presence of atrial septal defect (ASD), we referred the patient for CMR. Our differential diagnosis was for a significant left–right shunt. Chest X-ray finding in complete pericardial absence is known as “Snoopy sign” and include combination of levoposition of the heart, elongation and flattening of the left heart border, radiolucency between the pulmonary artery and aorta due to interposition of lung tissue and loss of the right heart border [7]. In the past, a diagnostic left pneumothorax has been used to confirm the diagnosis showing pneumopericardium [9]. Nowadays, the imaging modalities of choice for detecting pericardial absence are computed tomography and cardiac magnetic resonance. Since they are not affected by the acoustic window, they have a better sensitivity in identifying the levoposition of the heart and lung interposition in the aorto-pulmonary window and between the inferior cardiac border and the diaphragm. CMR is the gold standard for evaluating the cardiac anatomy, volumes and function and can demonstrate any functional abnormalities or regional bulging and herniation of the heart associated with pericardial agenesis. CT, on the other hand, has a greater spatial resolution and can be of help in identifying small defects [10,11].

Treatment depends on the type of the defect and clinical symptoms. Total and complete unilateral defects do not require treatment except for patients with significant symptoms. Partial defects in the symptomatic patient and in asymptomatic patients with signs of ventricular herniation should be corrected surgically [7]. Some authors suggest prophylactic closure of all small partial defects in order to prevent the risk of strangulation [12].

The patient’s history of a heart murmur in childhood, combined with echocardiographic findings suggesting a dilated right ventricle and an atypical heart position, necessitated further imaging studies, including CMR. Notably, the combination of a patent foramen ovale (PFO) and complete left-sided absence of the pericardium has not been previously documented in the scientific literature.

Concerns about a significant left-to-right shunt in the patient led us to this rare diagnosis. We believe that this clinical case demonstrates a practical approach to transitioning from a differential diagnosis to a definitive diagnosis. This insight can be of practical benefit to physicians and may advocate for the increased use of CMR in diagnosing congenital heart disease.

### Patient Perspective

The patient has a follow-up strategy once a year, with a follow-up visit after 6 months. The follow-up method will be transthoracic echocardiography. The risk of cryptogenic stroke is higher in patients with PFO than in the general population, the most prominent mechanism being paradoxical embolism. Closure of the atrial defect should be discussed with the patient. Since the patient is asymptomatic and the pericardial defect is large, no further treatment was recommended.

## 4. Conclusions

Congenital absence of the pericardium is a rare cardiac disorder. Although most of the cases are of little clinical significance, some of them are associated with serious complications, including risk of herniation and strangulation or coronary artery compression. Clinicians should be familiar with this anomaly because the early recognition can be lifesaving. Cardiac CT and CMR provide excellent anatomical and functional assessment and are the imaging modalities of choice for detecting pericardial defects. They help in selecting patients that would benefit from surgical repair.

## Figures and Tables

**Figure 1 reports-07-00048-f001:**
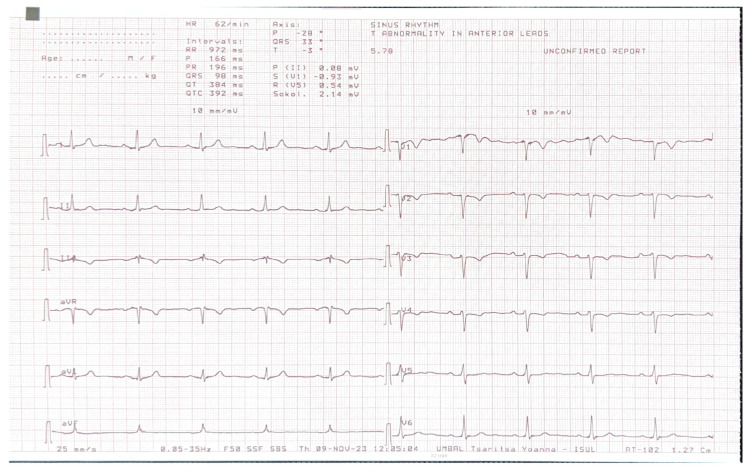
ECG of the patient.

**Figure 2 reports-07-00048-f002:**
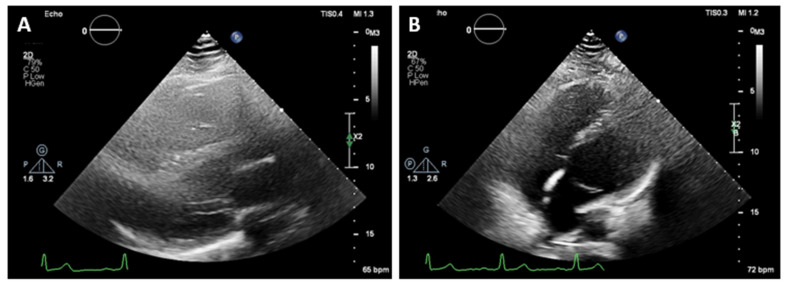
Two-dimensional transthoracic echocardiography. (**A**)—Parasternal long axis view gave the impression of right ventricular dilatation. Hyperechogenic space behind left ventricular posterior wall. Poor image quality. (**B**)—It was impossible to visualize the apex of the heart from the standard apical views in left lateral position.

**Figure 3 reports-07-00048-f003:**
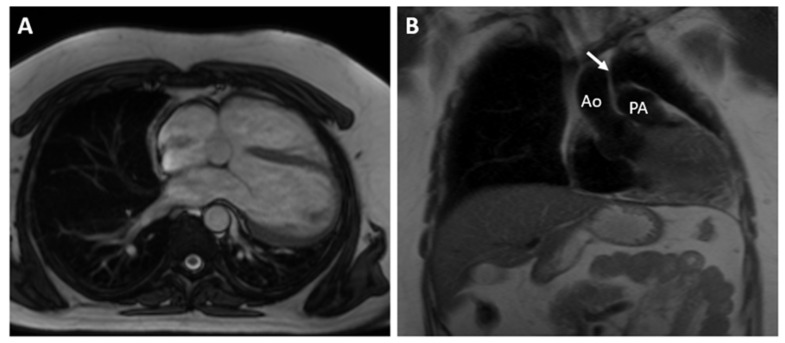
(**A**)—Axial image of the chest shows the displacement of the heart into the left hemithorax with the cardiac apex pointing laterally and posteriorly. (**B**)—Coronal image of the chest demonstrates the interposition of lung tissue between the aorta and pulmonary artery (white arrow). Ao—aorta; PA—pulmonary artery.

**Figure 4 reports-07-00048-f004:**
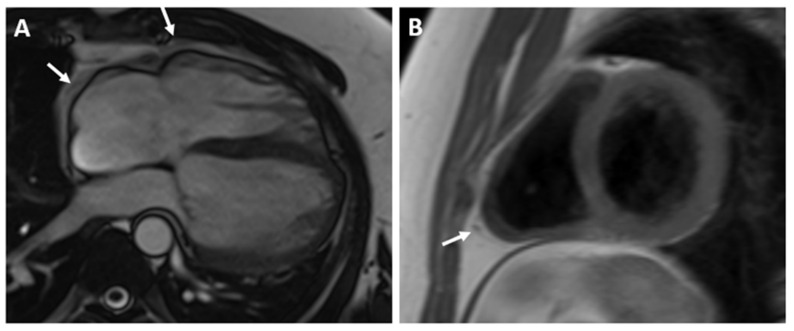
SSFP CINE image in four chamber view (**A**) and T1-weighted fast spin-echo in axial view (**B**) demonstrate discontinuous segments of pericardium surrounding the right atrium and right ventricular lateral wall. No detectable pericardium was found around the left ventricular wall.

## Data Availability

Data are contained within the article.

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
