# Peer review of "Complete Left-Sided Pericardial Congenital Absence"

_reports, 2024, doi:10.3390/reports7020048_

Round 1
Reviewer 1 Report
Comments and Suggestions for Authors
The article presented a case of Congenital Absence of Pericardium (CAP), a cardiac condition that is uncommon, and analyzed the of existing literature.
The article is well-structured, with sections for the introduction, case report, discussion, and conclusion.
Although there is a presentation of crucial components of CAP, there is a need to improve the discussion.
Please consider to include a paragraph on differential diagnosis in order to discuss other conditions that might mimic CAP, alerting clinicians in making accurate diagnoses.
Finally, please expand the management and long-term follow-up strategies, including potential complications and outcomes.
Author Response
Thank you for the constructive review.
We will try to improve and expand the discussion.
Differential diagnosis is important for clinical practice and we will try to present in more detail the diseases that could mimic this condition.
The patient has a follow-up strategy once a year, which we will describe in the article. He has a second follow-up visit in 6 months.
The potential risk is rather in the presence of PFO and we will try to discuss that as well.
Reviewer 2 Report
Comments and Suggestions for Authors
The author reports on a rare case of pericardial agenesis that did not require treatment. It is a well-written report with a very readable description. However, there is nothing novel about this case in terms of examination findings, complications, or treatment. The discussion is also limited to textbook descriptions, making it difficult to provide a new message to the reader.
Author Response
Cases of pericardial agenesis are rare, and the combination with PFO is even rarer.
The echocardiographic examination with an image resembling a dilated right ventricle and an atypical position of the heart prompts us to do additional imaging studies (CMR).
We believe that the clinical case shows practically how to reach the final diagnosis from the differential diagnosis. This would be of practical benefit to physicians and may increase the use of MRI in the diagnosis of congenital heart disease.
Reviewer 3 Report
Comments and Suggestions for Authors
I would like to congratulate the authors on the nice presentation of the interesting case. I have few comments,
- I understand the agensis descibes well the developmental abnormality of the absent pericardium. However, in other occasions in the text I think it is better to be consistent in using agenesis vs. absence, including the titel (may be absence better that agenesis). I have checked the ESC guidelines of the pericardial diseases from 2015, and it was only refered to as "absence" where agensis was never mentioned.
- Do you have also chest x ray?
- line 76 "Additionally, 75 patent foramen ovale was visualized with no significant left-to-right shunt (Qp/Qs = 1,1)". Is it really left to right? I think with a PFO it is right to left, if any. Otherwise it is an ASD or extended PFO.
- I was wondering if you planned any follow up echocardiography for the patient.
Kind regards,
Author Response
Thanks for your good reviews.
- In relation to the first comment, we use agenesis to emphasize the congenital absence of a pericardium. But we can change in part of the text by using them as synonyms.
- We did not do a chest X-ray. We didn't think it was necessary.
- When the shunt is hemodynamically insignificant, as in our case, it is described as left-to-right because of the greater pressures in the left atrium compared to those in the right. When the shunt becomes prominent and the dynamics change from right to left then it is described as a right-to-left shunt.
- The patient will be followed up once a year with transthoracic echocardiography. This has been explained to him and a second follow-up is due in 6 months.
Round 2
Reviewer 1 Report
Comments and Suggestions for Authors
The authors answered all my questions/suggestions.
Author Response
Thank you very much!
Reviewer 3 Report
Comments and Suggestions for Authors
I have no further comments, Kind regards,
Author Response
Thank you very much!